# Manual Therapy Facilitates Homeostatic Adaptation to Bone Microstructural Declines Induced by a Rat Model of Repetitive Forceful Task

**DOI:** 10.3390/ijms23126586

**Published:** 2022-06-13

**Authors:** Mary F. Barbe, Mamta Amin, Michele Y. Harris, Siva Tejaa Panibatla, Soroush Assari, Steven N. Popoff, Geoffrey M. Bove

**Affiliations:** 1Center for Translational Medicine, Lewis Katz School of Medicine, Temple University, Philadelphia, PA 19140, USA; mamta@temple.edu (M.A.); michele.harris@temple.edu (M.Y.H.); shivateja88@gmail.com (S.T.P.); gbove@bove.labs.com (G.M.B.); 2Department of Mechanical Engineering, College of Engineering, Temple University, Philadelphia, PA 19122, USA; soroush.assari@tempe.edu; 3Department of Biomedical Education, Lewis Katz School of Medicine, Temple University, Philadelphia, PA 19140, USA; spopoff@temple.edu; 4Bove Consulting and BoveLabs, Kennebunkport, ME 04046, USA

**Keywords:** overuse injury, work-related musculoskeletal disorders, massage therapy, repetitive motion disorder, osteoclasts, osteoblast

## Abstract

The effectiveness of manual therapy in reducing the catabolic effects of performing repetitive intensive force tasks on bones has not been reported. We examined if manual therapy could reduce radial bone microstructural declines in adult female Sprague–Dawley rats performing a 12-week high-repetition and high-force task, with or without simultaneous manual therapy to forelimbs. Additional rats were provided 6 weeks of rest after task cessation, with or without manual therapy. The control rats were untreated or received manual therapy for 12 weeks. The untreated TASK rats showed increased catabolic indices in the radius (decreased trabecular bone volume and numbers, increased osteoclasts in these trabeculae, and mid-diaphyseal cortical bone thinning) and increased serum CTX-1, TNF-α, and muscle macrophages. In contrast, the TASK rats receiving manual therapy showed increased radial bone anabolism (increased trabecular bone volume and osteoblast numbers, decreased osteoclast numbers, and increased mid-diaphyseal total area and periosteal perimeter) and increased serum TNF-α and muscle macrophages. Rest, with or without manual therapy, improved the trabecular thickness and mid-diaphyseal cortical bone attributes but not the mineral density. Thus, preventive manual therapy reduced the net radial bone catabolism by increasing osteogenesis, while rest, with or without manual therapy, was less effective.

## 1. Introduction

Musculoskeletal disorders secondary to occupational overuse are highly prevalent in many professions [1,2,3]. These injuries and disorders have profound effects on the neuromuscular system (which includes nerves, muscles, and tendons) [4]. Acute tissue injury results in acute inflammatory cell infiltration, while repeated injury is associated with chronic inflammation and fibrosis in tissues undergoing chronic injury/repair processes [5,6]. In humans, the risk factors for musculoskeletal disorders of the wrist include repetitive and sustained gripping, hand force, and combined exposure to both force and repetition [7,8].

We developed a rat model of upper extremity overuse injuries in which rats operantly perform an upper-extremity-reaching and lever-bar-pulling task at learned and defined reach rates and target forces for a food reward [9,10]. Young adult rats performing a high-repetition and high-force lever-pulling task for 12 weeks show 11% losses in trabecular bone volume in the distal radius and cortical bone thinning in the mid-diaphysis of the radius compared to the control rats [9,11,12,13]. The long term performance of this intensive lever-pulling task also results in muscle and systemic inflammatory responses and performance declines. Anti-inflammatory doses of ibuprofen for 8 weeks ameliorated the trabecular bone volume loss in the distal radius, as well as task-induced increases in serum inflammatory cytokines and bone inflammation and osteoclastogenesis and activity when provided in week 4 of a 12-week high-repetition and high-force task [11]. Similarly, reducing task demands in weeks 4–12 from this intensive task to an easier task (an ergonomic task reduction) also reduced the serum and muscle inflammatory responses and improved the trabecular bone volume in the distal radius.

Manual therapy (MT) is any therapy that is applied using the hands, including massage therapy and manipulative therapy. Several groups have examined the effects of various types of massage, manual therapy, or cyclical compression on functional abilities and on inflammation in soft tissues (neuromuscular tissues) in rodents and rabbits [14,15,16,17,18,19,20,21,22]. For example, we found that preventive manual therapy provided concurrently with task performance reduced inflammatory responses in nerve, muscle, and tendinous tissues, as well as somatosensory hypersensitivity, pathological neural discharge, and motor declines that develop with prolonged performance of intensive repetitive upper extremity tasks [19,21,22]. Yet, when manual therapy was provided with rest as a treatment after cessation following 12 weeks of task performance, the key neuromuscular changes were only partially improved. However, the effects of manual therapy as a treatment for task-induced bone degradation changes has not been examined to date. Several studies have examined potential changes in serum biomarkers of bone turnover in humans in response to massage and report increased levels of procollagen type 1 amino-terminal propeptide (P1NP) in postmenopausal women after Thai traditional massage [23,24] and increased levels of type I collagen C-terminal propeptide (PICP) in premature infants treated with a combination of massage and exercise [25], suggestive of increased bone formation in response to massage. Massage therapy has also been shown to improve bone mineralization and trabecular bone micro-architecture in growing juvenile and young adult rats [26]. The effects of manual therapy on the mature adult bone microstructure have not been examined.

Therefore, we sought here to examine for the first time whether manual therapy could reduce the catabolic effects of performing a high-repetition and high-force task for 12 weeks on the radius. Our protocol included gentle forearm tissue mobilization, forearm skin rolling, i.e., a treatment intended to emulate “myofascial release” or “muscle stripping”, to the forearm flexor compartment, mobilization of the wrist joints, and a gentle traction (stretch and glide) to the entire upper extremity. We also added direct manipulation of the palm in this study, where the tip of the rat’s index finger was pressed into the palm, at and just distal to the transverse carpal ligament, with a rolling motion. We hypothesized that our previously observed manual-therapy-induced reductions in muscle and systemic inflammatory responses may decrease osteoclast numbers and activity, a change that would reduce overall task-induced bone loss. We also sought to examine whether a six-week rest period after cessation of the task in week 12, with or without concurrent manual therapy, would improve bone microstructure or not. We hypothesized that rest would not be as beneficial as loading when performed during reduced inflammatory conditions yet were unsure if rest combined with manual therapy could enhance bone growth.

## 2. Results

### 2.1. Design

The design of this study is shown in Figure 1. The control groups were food-restricted 5%, similar to the TASK rats, but did not undergo task performance. One cohort of control rats received manual therapy (MT) to their upper limbs three times per week for 12 weeks (control + MT). The task rats were food-restricted to motivate them to work for a food pellet reward and were operantly shaped for 5–6 weeks to learn the high-force lever bar pulling. They went on to perform a high-repetition and high-force lever-pulling task for 12 weeks (see Methods). One cohort performed the task for 12 weeks untreated (TASK), while a second received manual therapy treatment of their upper limbs three times per week while continuing to perform the task for 12 weeks (TASK+MT). The last cohorts performed the task for 12 weeks before cessation of the task and then resting for 6 weeks untreated (TASK-R), or while receiving manual therapy treatment of their upper limbs three times per week (TASK-MTR).

The radial bone was examined in this study since we have previously shown in this model that the distal forearm bones are more affected by this task than the more proximal upper extremity bones (such as the humerus and scapula) [10] and that the radius is more affected than the ulnar bone [11,12].

### 2.2. Task-Induced Declines in Distal Radial Trabecular Microstructure Were Ameliorated by Preventive Manual Therapy

We first determined the effects of the 12-week TASK on the distal radial metaphyseal trabeculae microstructure using a micro-CT (Figure 2). Two-way ANOVAs revealed several task, treatment, and interaction differences (Table 1). Posthoc analyses were next performed. In the TASK rats, the microCT revealed several indices of catabolism, including decreased trabecular percent bone volume (% BV/TV), number (Tb.N), and bone mineral density (BMD) (Figure 2A,B,G), and increased trabecular separation (Tb.Sp), bone surface density (BS/BV), and degree of anisotropy (DA) compared to the controls (Figure 2C,E,F). The trabecular thickness (Tb.Th) was similar between the TASK and control rats (Figure 2D). Thus, the trabeculae in the distal radial bones of the TASK rats showed catabolic changes.

Providing 6 weeks of rest after TASK cessation (TASK-R rats) did not rescue the decreases in % BV/TV, Tb.N, or BMD (Figure 2A,B,G), nor increases in Tb.Sp. and DA compared to the controls (Figure 2C,F). Yet, the Tb.Th increased above the control levels (Figure 2D), and BS/BV decreased toward the control levels in the TASK-R rats compared to the TASK rats (Figure 2E). Thus, the task-induced declines in the distal radial trabecular microstructure were not reversed by rest alone.

We next examined the impact of manual therapy. There were no significant differences between the control and control+MT rats (Figure 2A–G). The TASK+MT rats had a similar trabecular microstructure to the control+MT rats, except for decreased Tb.Th and BMD (Figure 2D,G). Yet, the TASK+MT rats showed improvements in several aspects of trabecular microstructure compared to the TASK rats (increased % BV/TV, Tb.N, and BMD, and decreased Tb.Sp; Figure 2A,C,G). Thus, the task-induced declines in the distal radial trabecular microstructure were ameliorated by preventive manual therapy.

Six weeks of rest combined with manual therapy (TASK-MTR rats) resulted in continued decreases in BMD compared to the control+MT rats (Figure 2G) and no improvements in the trabecular microstructure compared to the TASK rats (Figure 2A–G). Although the TASK-MTR rats showed improved (higher) Tb.Th compared to the TASK+MT rats (Figure 2D), they also showed decreased Tb.N and BMD compared to the TASK+MT rats (Figure 2B,G). Thus, manual therapy provided after task cessation resulted in only mild improvements in the radial trabecular microstructure.

Figure 2H shows representative 3D transaxial and coronal models of the distal radial trabeculae from each group that match the graphical results shown in Figure 2A–G.

### 2.3. Preventive Manual Therapy Increased Osteoblast and Osteoid Indices in Radial Trabeculae

A static histomorphometry for osteoblastic indices was performed in distal metaphyseal trabeculae. The two-way ANOVAs revealed several task, treatment, and interaction differences (Table 1). The posthoc analyses showed that the TASK and TASK-R rats did not differ in osteoblast indices compared to the controls (Figure 3A–D). Yet, the TASK+MT rats showed increased osteoblast numbers (N.Ob/BS), osteoid volume (% OV/BV), osteoid surface (% OS/BS), and osteoid width (O.Wi) compared to the TASK rats (Figure 3A–D).

The TASK+MT rats also showed increased osteoblasts and osteoid volume compared to the control+MT rats (Figure 3A,B). These improvements were lost during the 6-week rest, with or without manual therapy (Figure 3A–C). Thus, preventive manual therapy increased the osteoblast and osteoid indices in the distal radial trabeculae compared to the TASK rats, improvements not observed in the rats receiving rest, with or without manual therapy.

### 2.4. Both Rest and Preventive Manual Therapy Reduced Osteoclast Surface and Numbers

A static histomorphometry for osteoclast parameters was performed. The two-way ANOVAs revealed differences by task, treatment, and their interaction (Table 1). The posthoc analyses showed the number of multinucleated TRAP+ or CD68+ osteoclasts per bone surface (N.Oc/B.S), and the percent of osteoclast surface to total bone surface ratio (% Oc.S/BS) values were considerably higher in the TASK rats compared to the controls (Figure 3E,F). Rest reduced these osteoclast parameters in the TASK-R rats compared to the TASK rats, as did manual therapy (i.e., TASK+MT rats) (Figure 3E,F). Rest after manual therapy treatment did not further alter the osteoclast parameters. Thus, manual therapy and rest treatments, alone or together, reduced the osteoclast numbers and percent surface.

### 2.5. Cortical Total Area and Periosteal Perimeter Improved with Preventive Manual Therapy

Since we have previously shown cortical thinning after prolonged performance of this high-repetition, high-force task [11,12,13], the cortical bone microstructure was examined using a microCT. The two-way ANOVAs primarily revealed task group differences (Table 1), so posthoc analyses were performed. The cortical bones in the radial mid-diaphyseal region showed microstructural changes in the TASK rats compared to the controls (Figure 4), including decreased total area (Tt.Ar), cortical bone area (Ct.Ar), percent Ct.Ar normalized to Tt.Ar (% Ct.Ar/Tt.Ar), cortical thickness (Ct.Th), cortical tissue mineral density (TMD) (Figure 4A–D,H), and increased marrow area (Ma.Ar) (Figure 4E). Thus, the cortical bone showed several catabolic changes in the TASK rats.

Yet, the TASK+MT rats showed increased Tt.Ar and periosteal perimeter (Ps.Pm), compared to the TASK rats (Figure 4A,F), and increased Ps.Pm, compared to the control+MT rats (Figure 4F). However, the TASK+MT rats had decreased Ct.Ar, % Ct.Ar/Tt.Ar, Ct.Th, and TMD, and increased M.Ar, compared to the control+MT rats (Figure 4B–E,G). Rest, with or without MT, improved most task-induced cortical structure declines, with the exception of TMD (Figure 4H). Figure 4G shows representative 3D transaxial models of the mid-diaphyseal region of the radius from each group. Thus, preventive manual therapy improved several cortical bone attributes in the TASK rats, as did rest, with or without manual therapy.

### 2.6. Preventive Manual Therapy Reduces Serum Biomarkers of Bone Resorption Activity and Inflammation

Two serum biomarkers of bone turnover were next assessed, osteocalcin (a serum biomarker of osteoblastic activity) and CTX-1 (C-telopeptide of type I collagen, a serum biomarker of osteoclast activity). The serum osteocalcin levels were lower in the TASK and TASK-R rats compared to the controls (Figure 5A) and higher in the TASK-MTR rats compared to the TASK-R and control+MT rats. The serum CTX-1 was higher in the TASK rats compared to the controls, TASK-R rats, and TASK+MT rats (Figure 5B). The serum TNF-α level was assessed due to its known contribution to increased osteoclastogenesis and activity [27]. The serum TNF-α levels were higher in the TASK rats compared to the control and TASK+MT rats (Figure 5C). See Table 1 for the ANOVA results. Thus, the TASK rats had increased serum biomarkers of bone degradation and inflammation but not osteoblastic activity.

### 2.7. Voluntary Task Parameters Improved with Preventive Therapy

To understand if muscle loading contributed to the observed radial bone changes, we examined voluntary task performance parameters in the TASK and TASK+MT rats. When the voluntary reach rate and pulling force on the lever bar were averaged across all weeks, the TASK+MT rats showed increases in both compared to the TASK rats (Figure 6A,B), yet they demonstrated decreased grasp duration (Figure 6C). Thus, the voluntary reach rate and pulling force improved with preventive manual therapy.

When these parameters were examined across weeks of performance (data for weeks 1, 3, 6, 9, and 12 are presented), group differences were observed (Table 1). Rats in each group had lower reach rates in most weeks than the target of four reaches/min. However, the TASK+MT rats neared that target at several points compared to the TASK rats (Figure 6D).

The TASK+MT rats were able to pull on the lever bar within the target range (160 to 200 cN) in all the weeks, while the TASK rats were not at several points (Figure 6E). While grasp duration showed significant time, treatment, and interaction effects (Table 1), no significant posthoc findings were observed (Figure 6F). All the rats were able to maintain the target grasp duration within the target range. Pearson’s rho tests showed a moderate positive correlation between the voluntary pulling force and trabecular percent bone volume, and a strong positive correlation between the voluntary pulling force and trabecular BMD (Table 2).

The estimated total volume of reaches across the 12 weeks was higher in the TASK+MT rats (Table 3) relative to the TASK rats, while the estimated total grasp duration on the lever bar was lower (Table 3). Overall, the total volume of loading (total reaches x total msec of grasping x the mean force per grasp) was higher in the TASK+MT versus TASK rats.

### 2.8. Muscle Inflammation Decreased with Preventive Manual Therapy and Correlated with Pulling Force

Since muscle inflammation is associated with reduced reach performance in this model [9,22,28], we next examined inflammatory macrophages (CD68+) in flexor digitorum muscles. See Table 1 for the ANOVA results. Increased numbers of CD68+ macrophages were observed in the TASK muscles compared to the controls (Figure 7A). This increase was ameliorated by both rest and manual therapy. Pearson’s rho tests showed moderate negative correlations between the muscle CD68+ macrophage numbers and voluntary pulling force (Figure 7B). Pearson’s rho tests showed moderate negative correlations between the CD68+ macrophage numbers and trabecular percent bone volume, trabecular BMD, and cortical total area (Table 2). Thus, muscle inflammation was reduced with preventive manual therapy and correlated with pulling force and several key attributes of radial bone microstructure.

No significant differences in the cross-sectional area of the entire flexor muscle of the TASK rats were observed compared to the controls (Figure 7C), although a significant increase was observed in the TASK-R rats compared to the controls. In contrast, both the TASK+MT and TASK-MTR rats showed increases in this muscle’s entire cross-sectional area compared to the control+MT rats. Pearson’s rho tests showed no correlation between this muscle’s cross-sectional areas and voluntary pulling force (Figure 7D), yet a low positive correlation with periosteal perimeter was observed (Table 2). Thus, the cross-sectional area of the entire flexor muscle was not a contributing factor to voluntary pulling force, although it may contribute to the size of the cortical periosteum perimeter.

### 2.9. Body Weight Was Not a Contributing Factor

Body weight was carefully controlled across the experiment to avoid body weight contributions to tissue catabolism. Consequently, the body weight did not differ across groups (Appendix A). Normalizing the percent bone volume to body weight did not change the earlier results (compare Appendix A to Figure 2A). Pearson’s rho tests showed no correlation of body weight to any trabecular or cortical bone attributes tested (Table 2).

## 3. Discussion

An involvement in heavy manual occupational tasks is linked to a higher incidence of neuro-musculotendinous injuries than nonmanual or light manual occupations [29,30]. A small number of studies have shown reduced bone mass [31,32] and bone and joint degradation in the wrists of humans involved in heavy or one-handed workloads [33]. Yet, the contributions of various stimuli generated by muscle on bone are still not fully understood [34,35,36]. We previously examined the effects of an operant 12-week lever-pulling task performed at high-repetition and high-force levels (defined as less than 30 s per cycle and over 31% of the maximum voluntary contraction, respectively [6,7]) on rat forelimb bones [9,11,37]. While we and others have shown that manual therapy treatment can improve the neuromuscular microstructure, to our knowledge, this is the first study examining the effect of manual therapy treatment on upper extremity bones undergoing overuse-injury-induced changes, and one of the most in-depth studies of the effects of manual therapy on mature adult bones. We also expanded our prior studies to now examine the effects of task cessation and a subsequent rest break of 6 weeks on the microstructure of the radius, with or without concurrent manual therapy treatment.

### 3.1. The 12-Week Upper-Extremity-Reaching-and-Grasping Task Enhanced Trabecular and Cortical Bone Catabolism

Similar to our past studies using this model and a similar task paradigm for 12 weeks [9,11,37], we found that the untreated TASK rats showed several indices of net trabecular bone catabolism in the distal radius compared to the control rats, including decreased trabecular bone volume, numbers, and bone mineral density, as well as increased trabecular separation, bone surface density, anisotropy, and osteoclast numbers. The TASK rats also showed several indices of catabolism in the radius’ mid-diaphyseal region, including cortical thinning and increased marrow narrowing, matching our past findings [11,37,38]. We and others have postulated that this net bone catabolism is a consequence of insufficient time to recover from mechanically induced microdamage and inflammatory responses accumulating from the prolonged static or cyclic overloading [9,36,39,40,41].

The increases in serum CTX-1 and TNF-α in the TASK rats help to support the above hypothesis. CTX-1 is a marker of the breakdown of the carboxyterminal telopeptide region of type I collagen by osteoclasts [42]. We have previously shown that the serum (and bone levels) of TNF-α increase significantly in rats performing a high-repetition and high-force task for 6 to 12 weeks compared to control rats [9,11,38]. When anti-inflammatory levels of ibuprofen were systemically administered, not only did the serum and bone levels of TNF-α decrease but so did the osteoclast numbers (Barbe et al., 2015; Jain et al., 2014). Systemic inflammatory cytokines can contribute to bone catabolism via their ability to activate osteoclastogenesis and osteoclast activity and subsequent bone resorption [43,44]. TNF-α levels have also been shown to inhibit osteoblast function and inhibit the osteocalcin gene promotor [44,45], further supporting a contribution of inflammation to bone turnover.

While it may seem a surprise to see osteoblast numbers at control levels in the trabeculae of 12-week TASK rats, they are increased in distal metaphyseal trabeculae immediately after the initial shaping period (TASK week 0) and in 3-week TASK rats compared to controls [46]. Apparently, short-term loading of this intensive task induces osteogenesis but not prolonged long-term loading. The low osteogenic response in week 12 may be due to the speculated equilibrium zone at which osteoblasts, and, therefore, bones, adapt to changes in their mechanical environment [47]. However, there is disagreement as to whether an equilibrium zone exists [48]. Perhaps, instead, the high circulating level of TNF-α is a contributing factor to the observed catabolism since it has been shown to negatively regulate bone formation by increasing osteoclast formation, differentiation, and activity, and to decrease osteoblast function [27,49].

### 3.2. Preventive Manual Therapy Improved Trabecular Microstructure and Partially Improved Cortical Bone Microstructure

We show here that manual therapy provided concurrently with long-term performance of an intensive lever-pulling task (i.e., TASK+MT rats) improves the radial trabecular microstructure compared to the TASK rats, with significant improvements in bone volume (greater), trabecular number (higher), trabecular separation (less), and bone mineral density (greater, although still lower than the control rats), osteoblast numbers (higher), and osteoclast numbers and osteoclast surface density (each lower). Although cortical bone thinning persisted in the TASK+MT rats, the total area and the periosteal perimeter were increased compared to the TASK rats. An increase in the periosteal perimeter, particularly in concert with an increase in the marrow area, is thought to be an adaptive response to cortical bone thinning to maintain resistance to bending [50,51].

Our trabecular microstructure findings are similar to those in a study by Chen and colleagues [26] showing an increased trabecular microstructure in response to massage therapy (assessed using histomorphometric methods). In their study, the rats received massage therapy for 10 min daily from postnatal day 6 through 10, with the bone then assessed up to postnatal day 60. This early-life massage therapy elicited prolonged anabolic effects on the bones. By postnatal day 60, when the rats were young adults, the trabecular thickness was greater in the male rats, while the trabecular separation was less in female rats. In additional, young adult rats that had received such early life massage showed greater bone mineral content (male rats) and increased bone mineralization in both sexes (in trabeculae in males and endosteum in females), suggesting that the anabolic processes stimulated by the massage during early development continued into at least young adulthood.

Studies examining adaptive bone formation suggest that brief daily periods of physiological loads can be strongly osteogenic [52,53]. While the mechanism of bone adaptation is not entirely clear, it is thought that mechanical strains sensed by bone are transduced into cellular signals. If the bone strain is under or over an “optimal strain” for a particular bone, then skeletal modeling or a remodeling response occurs [54,55]. While it is not feasible to calculate the specific loads that we provided to the bones, we can extrapolate the total loads applied during each session from force profiles gathered in a method used for training, which estimated force placed perpendicular to the forelimb using a 5-mm-diameter sensor [19,56]. Not including the skin rolling, we estimate that 150 N of force was delivered to each forelimb at each session. The direction of these forces favored bending the bones and shearing of the joints, which is distinct from the axial loading (or unloading) caused by the task. Since this treatment was a controlled variable, it seems likely that this amount and “shape” of loading are sufficient to help the bone maintain homeostatic osteogenesis in the face of otherwise damaging stimuli.

### 3.3. Rest, with or without Manual Therapy Treatment, Was Not as Effective a Preventive Manual Therapy in Rescuing Bone Microstructure

Six weeks of rest after task cessation, with or without manual therapy, did not improve the trabecular bone volume, number, separation, anisotropy, or bone mineral density compared to control rats. However, the rest did result in lowered osteoclast numbers and percent of osteoclast surface, changes that likely contributed to observed improvements in the trabecular thickness and bone surface density. In contrast, nearly all the attributes of cortical bone microstructure improved above the TASK levels (and back to control levels) after the 6 weeks of rest. Only the cortical tissue mineral density did not improve with rest. Perhaps, if the rest period had been longer, there would have been more tissue mineral density recovery, although the low osteoblast numbers in the resting animals suggests not.

### 3.4. Muscle Contributions

All the TASK rats, with or without manual therapy treatment, were able to meet the grasp duration requirements of holding the lever bar for at least 100 ms (and under 500 ms). However, across all weeks, the untreated TASK rats were not able to meet the target of four reaches per minute when also required to pull the lever bar at a minimum of 160cN of grasp force. The pulling force was negatively and moderately associated with the numbers of activated macrophages (CD68+) in the flexor digitorum muscles, the primary muscles used in performing this task. The TASK rats had increases in CD68+ macrophages in these muscles collected at 12 weeks in this study, and at 3, 6, or 12 weeks of task performance in past studies [21]. Muscle inflammatory responses reduce the grip strength and voluntary muscle performance [19,21,57,58]. In prior studies, we similarly observed that manual therapy reduces tissue inflammation [19,21,22] and task performance [21,22]. Interestingly, the pulling force correlated positively and strongly with the trabecular bone volume and BMD, while the numbers of CD68+ macrophages in the muscles correlated negatively and moderately with the trabecular bone volume and BMD, and negatively and strongly with the total area of the mid-diaphyseal cortical bone. These findings combined suggest that the manual therapy treatment improved the operant loading performance by decreasing the muscle and systemic inflammation, leading to both improved task performance and muscle loading on the underlying bones.

### 3.5. Limitations and Future Directions

There are some limitations in this study. We examined bones collected from only adult female rats to keep the study homogeneous (female rats are considerably smaller and weaker than male rats, for instance). Future studies should include males. The serum levels of osteocalcin and CTX-1 are representative of bone formation and resorption, respectively, occurring in all the bones involved in performing the task, not just the radius. Lastly, this study was performed in adult rats and, until a similar longitudinal study is performed in human subjects performing a similar repetitive high-force task, the translation of the findings should be made with discretion.

## 4. Materials and Methods

### 4.1. Experimental Design and Animals

All experiments were approved by the Temple University Institutional Animal Care and Use Committee (Temple University IACUC, protocol # 4850 from January 2019 until November 2021, and # 5120 from November 2021 until current) in compliance with NIH guidelines for the humane care and use of laboratory animals. Studies were begun on 56 young adult (2.5 months of age at onset) female Sprague–Dawley rats initially (Figure 1). Rats were housed individually in a central animal facility in a 12 h light: 12 h dark cycle with free access to water. All rats were first acclimated to the animal facility for 1 week. During the second week, the rats were handled and started food restriction to 5% less than historical data from age-matched rats with free access to food (used for weight comparison purposes only). This food restriction was necessary to motivate the rats to work for a food reward [59]. Thereafter, all rats were carefully maintained at the 5% food restriction for the duration of the experiment. Control rats used in this study were similarly food-restricted as rats performing the high-repetition and high-force lever-pulling task (see below), receiving similar amounts of rat chow and food reward pellets. All rats were weighed once or twice per week, provided regular rat chow daily in addition to food reward pellets (banana (F0024) and chocolate (F0165) grain-based dustless precision pellets, Bio-Serv, Flemington, NJ, USA), and allowed to gain weight over the course of the experiment, as previously shown for this model [11,19,60,61]. All rats were handled at least twice per week and provided cage enrichment toys that included chew bones, tunnels, and paper twists (Diamond Twists, Teklad 7979C.CS, Envigo, South Easton, MA, USA).

Rats were numbered and randomly assigned to groups by one group member (MA) at the beginning of the study to ensure blinding of tissue outcomes (operators providing the manual therapy treatments to specific rats could not be blinded; therefore, they were not engaged in the tissue analyses). Forty rats were randomly chosen to undergo an initial operant shaping period for 5–6 weeks to learn a high-force lever-pulling task (Figure 1) before then going on to perform a high-repetition and high-force lever-pulling task for 12 weeks. Task-performing rats were randomly divided into 4 subcohorts of n = 10/group: (1) rats that performed the task for 12 weeks untreated (TASK); (2) 12-week TASK rats that received upper extremity manual therapy concurrently, as described further below, while continuing to perform the task (TASK+MT); (3) 12-week TASK rats that ceased performing the task and then rested for 6 weeks (TASK-R); and (4) TASK rats that received manual therapy during the 6 weeks of rest (TASK-MTR). The remaining 17 rats were similarly acclimated, handled, and food-restricted, as described above, yet did not undergo operant shaping or task performance, without or with manual therapy during the same 12-week time frame, as the TASK rats (control, n = 10, and control+MT, n = 6). One each of the TASK+MT and TASK-MTR rats failed to learn the task and were excluded from the study, bringing the total number of rats down to 54 (Table 1).

### 4.2. Task Apparatus, Shaping, and Task Paradigm

Operant behavioral apparatuses that are linked to a computerized system that assays task outcomes and food reward dispensing were as previously described. In these operant chambers, forty rats were operantly shaped across a 6-week period in which they learned to perform a reaching and lever-pulling task at high-force loads (ramping upwards from naïve to pulling a lever bar at 50% of their maximum pulling force, which was 130 cN, for 10 min/day and 5 days/wk. During this shaping, the rats did not have to pull at a specified reach rate (the shaping methods used are as previously described [62]). The mean maximum voluntary pulling force of the rats was determined during shaping by requiring the rats to pull at increasing levels for a food reward until their maximum was reached; this maximum was 260 ± 7 cN (mean ± SEM). Rats went on to perform the high-repetition high-force task, also at 50% of their maximum pulling force, yet now with a target of 4 reaches/min for 2 h/day in 30 min intervals with 1.5 h rest breaks between 3 days/wk for 12 weeks [12]. This was performed using a custom-written software program that allowed us to choose a required force level for provision of a food reward (Force-Lever software, Med Associates, St. Albans, VT, USA), as previously described [12]. The primary limb used to reach was tracked each session by trained observers (the same individuals across all weeks).

### 4.3. Determination of Reach Performance Behaviors in the Task Rats

Task performance outcomes (mean number of reaches/minute and the voluntary pulling force on the lever bar) were recorded continuously by the computerized Force-Lever system during each task session and later extracted from the program data into executable file Excel spreadsheets. Data for each variable were generated by the program for each session for the TASK and TASK+MT rats and are presented as the mean per day and the mean of weeks 1, 3, 6, 9, and 12, the mean of all weeks together, or as the estimate of totals across all weeks. The TASK-R rats performed the same task as the TASK rats; data cannot be reported for their final 6 weeks as they were resting. These data could not be generated for control rats as they did not perform the task and were resting.

### 4.4. Manual Therapy

The treatment was based on a previously developed manual therapy protocol performed on unsedated rats [22]. It included gentle forearm tissue mobilization, forearm skin rolling, i.e., a treatment intended to emulate “myofascial release” or “muscle stripping”, to the forearm flexor compartment, mobilization of the wrist joints, and a gentle traction (stretch and glide) to the entire upper extremity. We also added direct manipulation of the palm in this study, where the tip of the rat’s index finger was pressed into the palm at and just distal to the transverse carpal ligament with a rolling motion. The individuals providing these treatments were the same as in previous studies [19,20]. Treatments to TASK-MT rats were performed 3 times per week on alternate days to the task performance for 7 weeks and lasted approximately five to seven minutes per limb, i.e., a total of 10–14 min per rat. TASK-MTR rats were treated similarly during the 6-week rest period.

### 4.5. Tissue Collection

Animals were deeply anesthetized with 5% isoflurane in oxygen and then euthanized by performing thoracotomy and cardiac puncture for blood collection using a 18-gauge needle. Collected blood was placed into uncoated 15 mL tubes and stored on ice for 1 h before being centrifuged at 12,000 rpm for 20 min at 4 °C. Serum was harvested and frozen at −80 °C until assayed. After blood collection, all rats received a transcardial perfusion of first 0.9% saline and then 4% paraformaldehyde in 0.1 M PO_4_ buffer, pH 7.4. Forelimb bones and flexor digitorum muscles (superficialis and profundus muscles are combined in rats) were collected from the primary reach limbs of all task rats and from one limb of each control rat using previously described methods [19]. Collected tissues were then immersion-fixed for 48 h in the buffered 4% paraformaldehyde fixative. Euthanasia and tissue collection were performed by one person throughout the study; this person was naïve to group assignment.

### 4.6. MicroCT Imaging and Analysis

After fixation, collected forelimb bones were stored in phosphate buffered saline (PBS) with sodium azide until they underwent microCT scanning using a Skyscan 1172 (Skyscan, Microphotonics, Allentown, PA, USA, a subsidiary of Bruker, Kontich, Belgium), 12 megapixel high-resolution cone-beam microCT scanner (Bruker, Kontich, Belgium) using the following settings: a 5.89 μm isotropic voxel size, X-ray source spot size of 300 nm, Al 0.5 mm filter, voltage of 62 kV, current of 167 μA, rotation step of 0.40°, frame averaging of 5. Reconstruction of scanned images was performed using a ring artifact correction of 10 and a beam hardening correction of 60% using Bruker NRecon software. Skyscan volume rendering software (CTVox) and analysis software (CTAn) were used to render 3D models and data, respectively (Skyscan, Microphotonics, Allentown, PA, USA, a subsidiary of Bruker, Kontich, Belgium). MicroCT analysis was performed, as previously described [62], in metaphyseal and diaphyseal regions of the radius by an individual who was blinded to group assignment. We focused on analyzing radial bone changes as we have previously determined that the radius undergoes considerable task-induced changes [11,12]. Briefly, 50 image slices were skipped proximally from the end of the distal growth plate. Then, regions of interest of image slices were drawn a few pixels inwards from endocortical boundary to segment out a volume of interest of trabecular bone for 500 image slices. These images were analyzed using a lower threshold of 75 and an upper threshold of 255 to separate trabecular bone versus non-bone.

MicroCT analysis was also performed of the cortical bone in the mid-diaphyseal region of the radius, as previously described [12]. The cortical diaphyseal regions of interest were 7 to 9 mm from the end of the distal growth plate of the radius (approximately 45 to 55% length of total radius bone). These images were analyzed using a lower threshold of 108 and an upper threshold of 255 to separate cortical bone versus non-bone. Cortical bone morphometric traits were computed from binarized images using 2D techniques. Trabecular volumetric bone mineral density (BMD) and cortical bone tissue mineral density (TMD) was determined after calibration using two 4-mm calcium hydroxyapatite phantoms with densities of 0.25 g cm^3^ and 0.75 g cm^3^ (Bruker, Kontich, Belgium).

### 4.7. Bone Histomorphometry

After microCT analysis, the forearm bones were processed for histomorphometry. Subcohorts of bones (n = 3–5/gp) were processed and embedded in methyl methacrylate resin and sectioned into 5 μm longitudinal sections and placed on charged slides (embedding and sectioning was performed by Bioquant Image Analysis Incorporation, Nashville, TN, USA). Slides were either left unstained or were stained with Masson’s Trichrome or Safranin O/Fast Green. The remaining forelimb bones of all preferred reach limbs were paraffin-embedded after a 6-week decalcification period (Cat # NC9044643, StatLab Immunocal solution, Fisher Scientific, Pittsburgh, PA, USA). For this, forelimb bones were embedded in separate paraffin blocks, sectioned longitudinally into 5-μm-thick sections, and placed onto charged slides. These sections were dried onto charged slides using slide warming plates overnight before being stored at room temperature until use, at which point they were deparaffinized in xylene and decreasing concentrations of ethanols before staining. Sections with radial bone underwent Masson’s Trichrome or Safranin O/Fast Green staining or immunohistochemical staining with either an anti-TRAP or CD68+ antibody using previously described methods [10,46]. Specificity of the TRAP antibody has been previously described, as has the detection of TRAP+ cells by CD68 immunostaining [10,46]. Only multinucleated TRAP+ or CD68+ cells were counted with the bone to quantify osteoclasts.

Bone histomorphometry was performed by one individual using a semi-automated image analysis software (BIOQUANT Osteo, Bioquant, Nashville, TN, USA) that was interfaced with a Nikon E800 microscope (Nikon, Melville, NY, USA) and a digital camera (Retiga 4000R QImaging Firewire Cameras, Surry, BC, Canada). Coded slides were used to mask group assignment. Static histomorphometry parameters of trabecular bone microarchitecture were assayed using both plastic- and paraffin-embedded sections in Masson’s Trichrome stained sections for osteoblast parameters and in TRAP- or CD68+-stained sections for osteoclast parameters. Trabeculae were assayed in the distal metaphysis beginning 150 μm below the chondro-osseous junction of the secondary spongiosa and 50 μm in from the surrounding cortical bone using a 20× objective; at least 2 sections were quantified per rat bone and using published guidelines [63,64].

### 4.8. Muscle Histomorphometry

After intracardial perfusion with fixative followed by immersion fixation, a 2.5-mm-thick piece of flexor digitorum muscle was removed from mid- to proximal forearm and then cryopreserved first in 10% and then in 30% sucrose in phosphate buffer across 4 days (48 h per sucrose solution). Tissues were embedded in Optimum Cutting Temperature compound (23730571, FisherScientific, Houston, TX, USA) and cryosectioned at 14 μm (cross-sectionally for the proximal to mid-flexor digitorum muscle mass, and longitudinally for the remaining distal muscle mass). Sections were placed onto charged slides (22037200, FisherScientific, Pittsburgh, PA, USA) and dried overnight at room temperature before storage in foil-wrapped slide boxes at −80 °C until use.

Subsets of muscle cryosections were either stained with hematoxylin and eosin or in a batched set with a specific antibody against CD68 (a marker of primarily activated type macrophages in rats [65,66], ab31630, Abcam) at 1:300 dilution in PBS. Antibodies were incubated with the sections overnight at room temperature (after pepsin digestion using a 1:6 dilution of 12,000 units/mL of pepsin in 0.01 N HCL for 15 min, and then a 30 min incubation in 10% goat serum). Sections were then washed in PBS (3 times, 5 min each). This was followed by incubation with an appropriate secondary antibody conjugated to a red fluorescent tag (Cy3) (Jackson ImmunoResearch, West Grove, PA, USA) at a dilution of 1:100 for 2 h at room temperature. DAPI was used as a nuclear counterstain following the immunostaining. Each subset was coverslipped with 80% glycerol in phosphate buffer to prevent tissue shrinkage.

Whole muscle cross-sectional area was analyzed in hematoxylin- and eosin (H&E) -stained sections performed by one individual using the image analysis system described above. Again, coded slides were used to mask group assignment. The widest circumference (in μm) of the flexor digitorum muscles was quantified (in μm^2^) from these cross-sections at 10× magnification. Numbers of CD68 cells within the muscles were counted in the mid-muscle cross-sections and in longitudinally cut sections at myotendinous junctions in three randomly chosen fields per muscle region and per rat in at least 3 cross-sections from the mid-muscle mass per rat limb.

### 4.9. Serum ELISAs

Serum samples were batch-assayed for: (1) osteocalcin (a serum biomarker of osteoblastic activity [67,68,69], AC-12F1, Rat-MID Osteocalcin EIA, Immunodiagnostic Systems, Tyne & Wear, East Boldon, Boldon Colliery, UK); (2) CTX-1 (C-telopeptide of type I collagen, a serum biomarker of bone resorption, AC-06F1, RatLaps EIA, Immunodiagnostic Systems); and (3) TNF-α (tumor necrosis factor alpha, a key pro-inflammatory cytokine, EA100366, Origine, Rockville, MD). ELISAs were conducted using manufacturers’ protocols, in duplicate, by one individual who was naïve to group assignments.

### 4.10. Statistical Analyses

GraphPad PRISM version 9 was used for the statistical analyses. The sample size for this study was derived from our previous studies [20,22], assuming a power of 80% and level of significance of 0.05, and a standard deviation of 28 for grip strength. All data are expressed as mean ± 95% confidence intervals (CIs). Further, *p* values of <0.05 were considered significant for all comparisons. Two-way ANOVAs were used to compare difference in microCT, histological, muscle, and serum findings using the factors task (control, TASK, versus TASK-R) and treatment (untreated versus manual therapy treated), followed by Sidak multiple comparison posthoc tests in which *p*-values were adjusted for the multiple comparisons. Voluntary task parameters (reach rate, pulling force, and grasp duration) of all weeks together for the TASK and TASK+MT groups were compared using unpaired two-tailed *t*-tests. Voluntary task parameters across time for the TASK and TASK+MT groups were compared using a repeated measures mixed-effect model (REML, REstricted Maximum Likelihood model) and the factors of time and treatment, followed by Sidak multiple comparison posthoc tests. Adjusted *p* values are reported. Pearson’s rho tests were used to perform correlational assays. Values between 0.4 and 0.59 (−0.4 and −0.59) were considered as moderately positive (or negative) relationships, and values between 0.6 and 0.79 1.0 (−0.7 and −1.0) as strongly positive (or negative) relationships, and 0.8–1.0 as very strong correlations [70,71].

## 5. Conclusions

Rats performing a 12-week intensive, high-repetition, and high-force task demonstrated increased indices associated with bone resorption (catabolic), leading to a net decrease in trabecular bone mass and cortical thinning and reduced density in each region examined. Preventive manual therapy, provided three times per week during the 12 weeks of task performance, reduced the catabolic changes in the distal radial metaphyseal region, reduced osteoclasts, as well as reduced muscle and systemic inflammatory responses and enhanced osteoblast indices, maintaining the bone microstructure at control levels. However, the preventive manual therapy did not prevent the cortical microstructure changes. Rest, with or without manual therapy treatment, effectively reduced the osteoclast numbers, improved the trabecular thickness in the distal radial metaphyseal region, and restored the cortical microstructure to control levels. However, none of the interventions used restored the task-induced declines in the trabecular bone mineral density or cortical bone tissue mineral density to control levels.

## Figures and Tables

**Figure 1 ijms-23-06586-f001:**
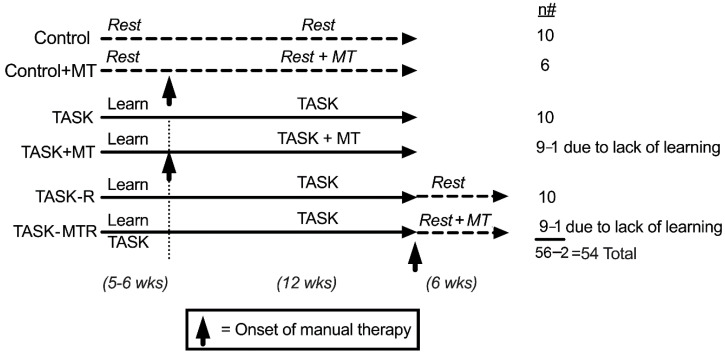
Experimental design. Abbreviations: control rats (Control); control rats that received manual therapy for 12 weeks (Control+MT); 12-week TASK rats (TASK); TASK rats that simultaneously received manual therapy for 12 weeks (TASK+MT); TASK rats that rested for 6 weeks after task cessation (TASK-R), TASK rats that rested for 6 weeks after task cessation while receiving manual therapy treatment of their upper limbs three times per week (TASK-MTR). n# = number of rats per group; wks = weeks.

**Figure 2 ijms-23-06586-f002:**
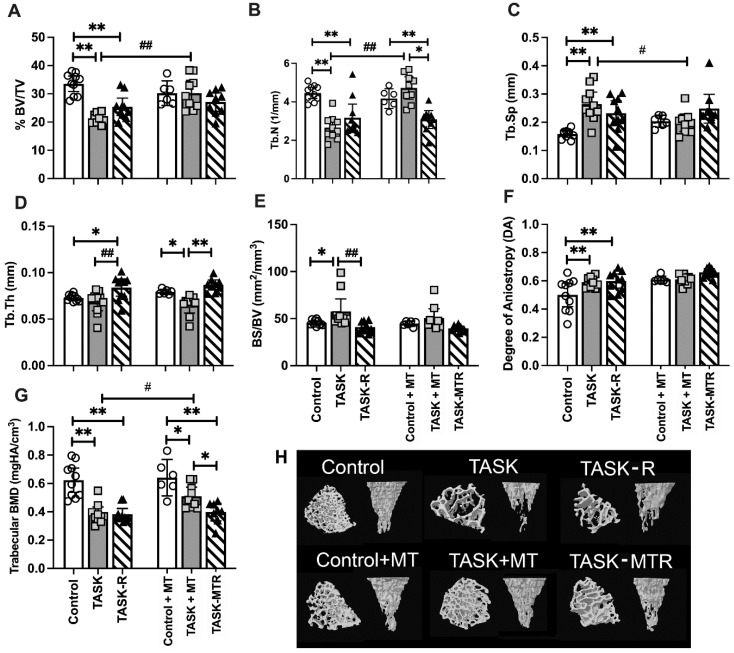
Micro-computerized tomography (MicroCT) of distal radial metaphyseal trabecular bone. (**A**–**G**) Percent bone volume normalized to total volume (% BV/TV), trabecular number (Tb.N), trabecular separation (Tb.Sp), trabecular thickness (Tb.Th), bone surface density (bone surface normalized to bone volume; BS/BV), degree of anisotropy (DA), trabecular bone mineral density (BMD). Mean ± 95% confidence intervals (CIs) shown. (**H**) Representative transaxial and coronal images of the middle region of the trabecular zone of the distal radial metaphysis for each group. The transaxial view is shown from the top near the growth plate. Abbreviations: control rats (Control); 12-week TASK rats (TASK); TASK rats that rested for 6 weeks after task cessation (TASK-R); control rats that received manual therapy for 12 weeks (Control+MT); TASK rats that simultaneously received manual therapy for 12 weeks (TASK+MT); TASK rats that rested for 6 weeks after task cessation while receiving manual therapy treatment of their upper limbs three times per week (TASK-MTR). # and ##: *p* < 0.05 and *p* < 0.01, respectively, compared to TASK group. * and **: *p* < 0.05 and *p* < 0.01, respectively, compared between groups as shown.

**Figure 3 ijms-23-06586-f003:**
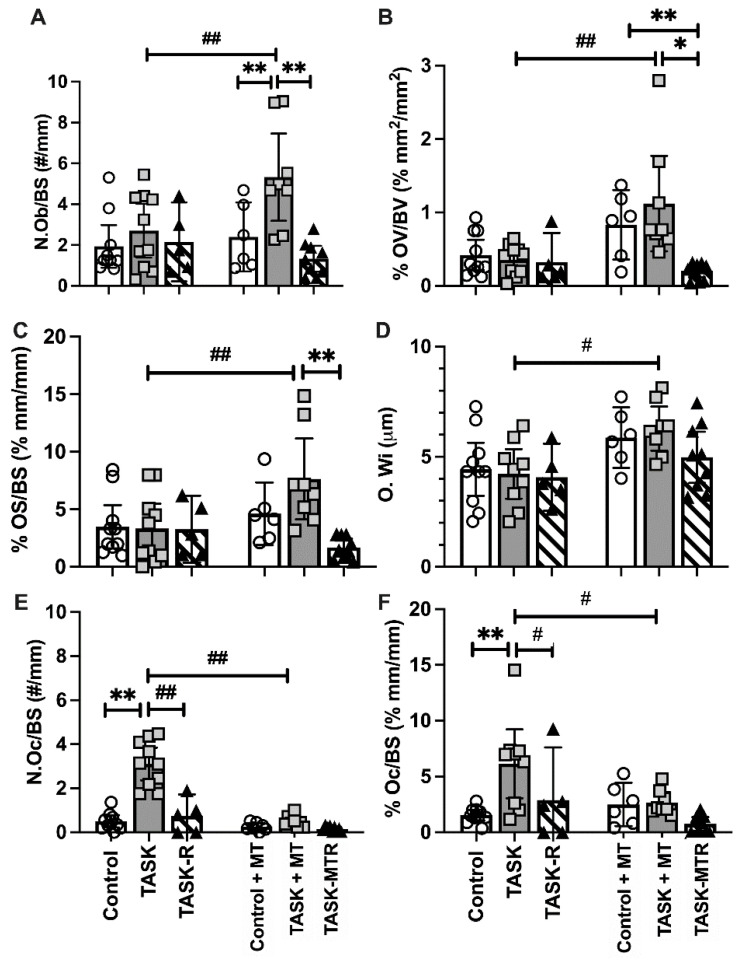
Static histomorphometry of osteoblast and osteoclast parameters in the distal radial metaphyseal trabecular bone. This was performed in a region located 150 μm below the chondro-osseous junction of the secondary spongiosa and 50 μm in from the surrounding cortical bone. (**A**–**F**) Number of osteoblasts per bone surface (N.Ob/BS), percent osteoid volume normalized to bone volume (% OV/BV), percent osteoid surface normalized to bone surface (% OS/BS), osteoid width (O.Wi), number of osteoclasts per bone surface (N.Oc/BS), percent osteoclast surface normalized to bone surface (% Oc.S/BS). Mean ± 95% confidence intervals (CIs) shown. Abbreviations: control rats (Control); 12-week TASK rats (TASK); TASK rats that rested for 6 weeks after task cessation (TASK-R); control rats that received manual therapy for 12 weeks (Control+MT); TASK rats that simultaneously received manual therapy for 12 weeks (TASK+MT); TASK rats that rested for 6 weeks after task cessation while receiving manual therapy treatment of their upper limbs three times per week (TASK-MTR). # and ##: *p* < 0.05 and *p* < 0.01, respectively, compared to the TASK group. * and **: *p* < 0.05 and *p* < 0.01, respectively, compared between groups as shown.

**Figure 4 ijms-23-06586-f004:**
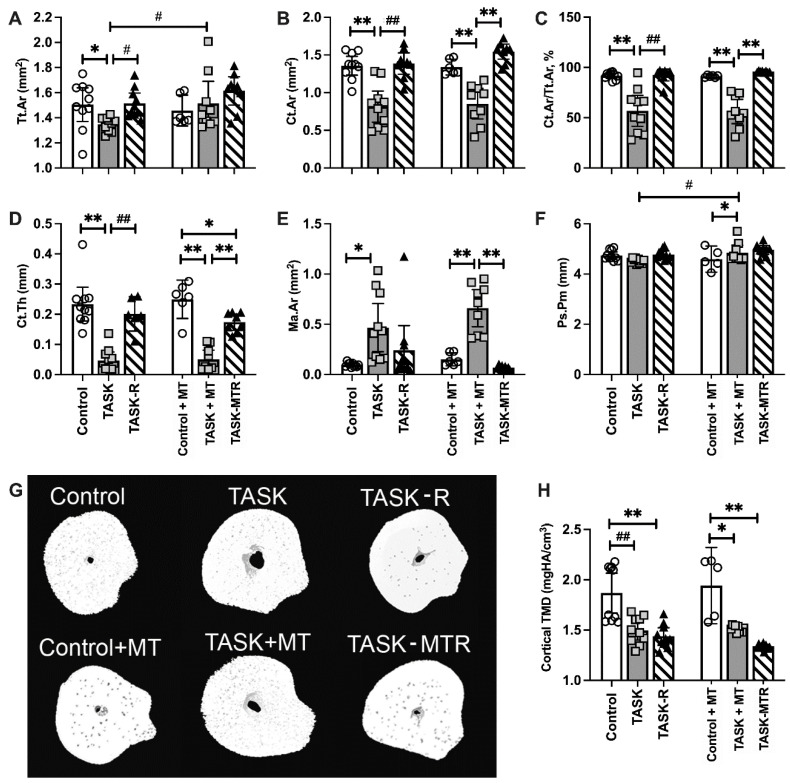
MicroCT of the mid-diaphyseal cortical bone of the radius. (**A**–**F**) Total area (Tt.Ar), cortical bone area (Ct.Ar), percent cortical area normalized to total area (% Ct.Ar/Tt.Ar), cortical thickness (Ct.Th), marrow area (Ma.Ar), periosteal perimeter (Ps.Pm). (**G**) Representative transaxial images of the mid-diaphysis of the radius. (**H**) Cortical bone tissue mineral density (TMD). Mean ± 95% confidence intervals (CIs) shown. Abbreviations: control rats (Control); 12-week TASK rats (TASK); TASK rats that rested for 6 weeks after task cessation (TASK-R); control rats that received manual therapy for 12 weeks (Control+MT); TASK rats that simultaneously received manual therapy for 12 weeks (TASK+MT); TASK rats that rested for 6 weeks after task cessation while receiving manual therapy treatment of their upper limbs three times per week (TASK-MTR). # and ##: *p* < 0.05 and *p* < 0.01, respectively, compared to the TASK group. * and **: *p* < 0.05 and *p* < 0.01, respectively, compared between groups as shown.

**Figure 5 ijms-23-06586-f005:**
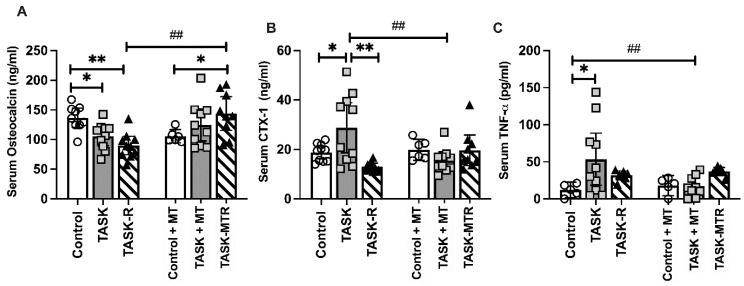
Serum levels of bone turnover and inflammation biomarkers. (**A**) Osteocalcin. (**B**) C-telopeptide of type I collagen (CTX-1). (**C**) Tumor necrosis factor-alpha (TNF-α). Abbreviations: control rats (Control); 12-week TASK rats (TASK); TASK rats that rested for 6 weeks after task cessation (TASK-R); control rats that received manual therapy for 12 weeks (Control+MT); TASK rats that simultaneously received manual therapy for 12 weeks (TASK+MT); TASK rats that rested for 6 weeks after task cessation while receiving manual therapy treatment of their upper limbs three times per week (TASK-MTR). Mean ± 95% confidence intervals (CIs) shown. ##: *p* < 0.01, compared to the TASK group; * and **: *p* < 0.05 and *p* < 0.01, respectively, compared between groups as shown.

**Figure 6 ijms-23-06586-f006:**
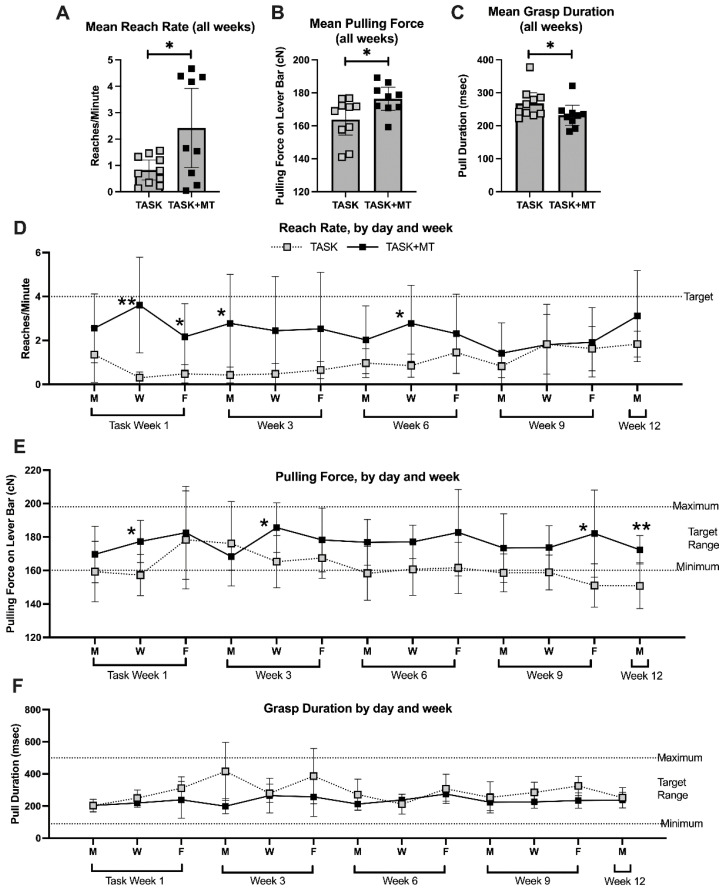
Voluntary task outcomes. Data are shown only for rats performing the task immediately before the time of tissue collection (TASK and TASK+MT). (**A**) Mean reach rate (mean reaches per minute) across weeks. (**B**) Mean pulling force on the lever bar across weeks. (**C**) Mean grasp duration (mean time rat spent holding the lever bar) across weeks. (**D**–**F**) Mean reach rate, mean pulling force, and mean grasp duration, respectively, shown for selected days (the four sessions of each day were averaged) and weeks (1, 3, 6, 9, and 12). Abbreviations: 12-week TASK rats (TASK); TASK rats that simultaneously received manual therapy for 12 weeks (TASK+MT). Mean ± 95% confidence intervals (CIs) shown. * and **: *p* < 0.05 and *p* < 0.01, respectively, compared to TASK group.

**Figure 7 ijms-23-06586-f007:**
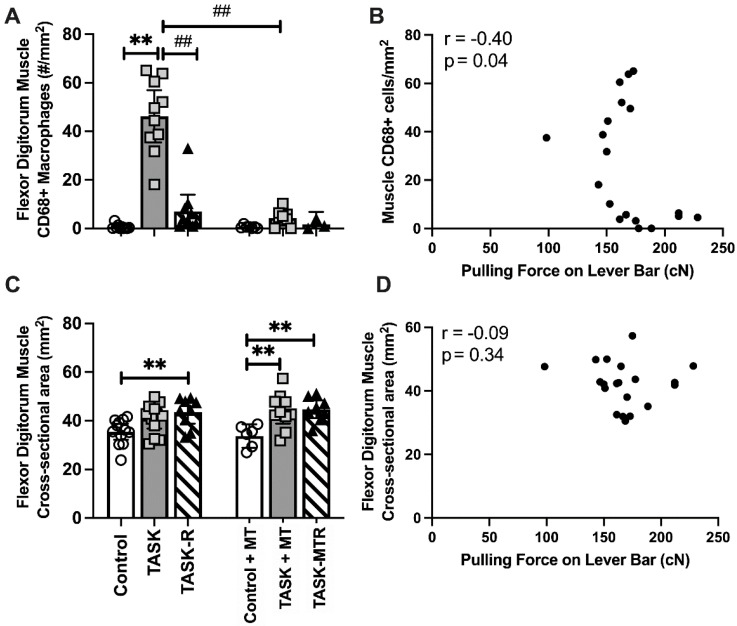
Assays of flexor digitorum muscle and Pearson’s correlations. (**A**) Number (#) of CD68 immunopositive macrophages in muscle. (**B**) Pearson’s correlation between # CD68+ cells in muscle and mean pulling force during the final two weeks of task performance. (**C**) Cross-section area (CSA) of flexor digitorum muscle at its widest point in mid-forearm. (**D**) Pearson’s correlation between the muscle CSA and mean pulling force during the final two weeks of task performance. Mean ± 95% confidence intervals (CIs) shown. Abbreviations: control rats (Control); 12-week TASK rats (TASK); TASK rats that rested for 6 weeks after task cessation (TASK-R); control rats that received manual therapy for 12 weeks (Control+MT); TASK rats that simultaneously received manual therapy for 12 weeks (TASK+MT); TASK rats that rested for 6 weeks after task cessation while receiving manual therapy treatment of their upper limbs three times per week (TASK-MTR). ##: *p* < 0.01, compared to the TASK group; **: *p* < 0.01, compared between groups as shown.

**Table 1 ijms-23-06586-t001:** Two-way ANOVA or mixed model results. Significant findings are bolded.

Attribute	Task Group	Treatment Group	Interaction
Metaphyseal Trabeculae, MicroCT (2-way ANOVA)			
% BV/TV	***p* = 0.0001**	***p* = 0.04**	***p* = 0.0004**
Tb.N	***p* < 0.0001**	***p* = 0.004**	***p* < 0.0001**
Tb.Sp	***p* = 0.003**	*p* = 0.91	***p* = 0.005**
Tb.Th	***p* < 0.0001**	*p* = 0.42	*p* = 0.32
BS/BV	***p* = 0.0004**	*p* = 0.27	*p* = 0.59
DA	***p* = 0.006**	***p* = 0.001**	*p* = 0.191
BMD	***p* < 0.0001**	***p* = 0.04**	*p* = 0.12
Metaphyseal Trabeculae, Histomorphometry (2-way ANOVA)			
N.Ob/BS	***p* = 0.002**	*p* = 0.14	***p* = 0.03**
% OV/BV	***p* = 0.008**	***p* = 0.006**	***p* = 0.02**
% OS/BS	***p* = 0.02**	*p* = 0.14	***p* = 0.02**
O.Wi	*p* = 0.36	***p* = 0.002**	*p* = 0.55
N.Oc/BS	***p* < 0.0001**	***p* < 0.0001**	***p* < 0.0001**
% Oc/BS	***p* = 0.005**	***p* = 0.03**	***p* = 0.03**
Mid-diaphyseal cortical bone, MicroCT (2-way ANOVA)			
Tr.Ar	***p* = 0.03**	*p* = 0.09	*p* = 0.14
Ct.Ar	***p* < 0.0001**	*p* = 0.31	*p* = 0.45
Ct.Ar/Tr.Ar, %	***p* < 0.0001**	*p* = 0.83	*p* = 0.85
Ct.Th	***p* < 0.0001**	*p* = 0.93	*p* = 0.38
Ma.Ar	***p* < 0.0001**	*p* = 0.71	*p* = 0.056
Ps.Pm	***p* = 0.04**	*p* = 0.10	***p* = 0.04**
TMD	***p* < 0.0001**	*p* = 0.94	*p* = 0.32
Serum Biomarkers (2-way ANOVA)			
Osteocalcin	*p* = 0.81	*p* = 0.08	***p* = 0.0003**
CTX-1	*p* = 0.06	*p* = 0.31	***p* = 0.0005**
TNF-α	*p* = 0.12	*p* = 0.27	***p* = 0.04**
Flexor Digitorum Muscle (2-way ANOVA)			
# CD68+ cells	***p* < 0.0001**	***p* < 0.0001**	***p* < 0.0001**
CSA (whole muscle)	***p* < 0.0001**	*p* = 0.57	*p* = 0.44
**Attribute**	**Time**	**Treatment**	**Interaction**
Voluntary Reach Outcomes (Mixed Model)			
Reach Rate	*p* = 0.28	***p* = 0.04**	***p* = 0.01**
Pulling Force	*p* = 0.34	***p* = 0.01**	*p* = 0.49
Grasp Duration	***p* = 0.047**	***p* = 0.04**	***p* = 0.03**

Abbreviations as indicated in the text of the results.

**Table 2 ijms-23-06586-t002:** Correlations of behavior and muscle features with key radial bone trabecular and cortical bone microstructural attributes. Significant findings are bolded.

Behavior or Muscle Feature	Tb. % BV/TV	Tb. BMD	Ct. Tr.Ar.	Ct. Ps.Pm	Ct TMD
Reach Rate (reaches/min)	r = 0.16 *p* = 0.25	r = 0.04 *p* = 0.42	r = 0.04 *p* = 0.43	r = 0.09 *p* = 0.43	r = 0.03 *p* = 0.44
Pulling Force (cN)	**r = 0.53** ***p* = 0.009**	**r = 0.63** ***p* = 0.002**	r = 0.06 *p* = 0.39	r = 0.12 *p* = 0.30	r = −0.10 *p* = 0.33
Grasp Duration (msec)	r = −0.35 *p* = 0.13	r = −0.33 *p* = 0.17	r = −0.40 *p* = 0.09	r = −0.36 *p* = 0.13	r = 0.04 *p* = 0.85
Flexor Muscle CD68+ cells	**r = −0.47** ***p* = 0.03**	**r = −0.46** ***p* = 0.04**	**r = −0.55** ***p* < 0.0001**	r = 0.01 *p* = 0.94	r = 0.16 *p* = 0.28
Flexor Muscle CSA (mm^2^)	r = −0.01 *p* = 0.94	r = 0.22 *p* = 0.24	r = 0.03 *p* = 0.79	**r = 0.28** ***p* = 0.03**	r = 0.10 *p* = 0.58
Body Weight	r = −0.19 *p* = 0.18	r = 0.14 *p* = 0.30	r = −0.14 *p* = 0.19	r = −0.11 *p* = 0.25	r = −0.20 *p* = 0.20

Ct = mid-diaphyseal cortical bone; Tb = trabecular bone. Other abbreviations as indicated in the text of the results.

**Table 3 ijms-23-06586-t003:** Estimated total volume of high-repetition, high-force loading across 12 weeks.

Reach Parameter	TASK	TASK+MT
	Mean ± SEM	Mean ± SEM
Total volume of reaches (total reaches)	315 ± 4	739 ± 5
Mean pulling force loads (cN)	161 ± 4	176 ± 3
Total grasp duration of lever bar (s)	100.1 ± 0.03	65 ± 0.1
Total volume of loading (total reaches · total ms grasping · mean force per grasp (kNs))	50.7 ± 2	84.5 ± 2

## Data Availability

The authors confirm that the data supporting the findings of this study are available within the article and/or its Appendix A.

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
