# Peer review of "Manual Therapy Facilitates Homeostatic Adaptation to Bone Microstructural Declines Induced by a Rat Model of Repetitive Forceful Task"

_ijms, 2022, doi:10.3390/ijms23126586_

Round 1

Reviewer 1 Report

Abstract: Please do not start with what kind of model you have, but with a general overview of the field you are working in.

The first paragraphs of the introduction should rather focus on what is already known on manual therapy and its impact on bone metabolism instead of referencing your own works.

Results

The title of the first paragraph should not include a reference to the figure.

Where is the main text to 2.1?

Why was the radius chosen?

Tab. 1: Rather than the p-values, the means and standard deviation should be shown.

Without further explanation or verification, Fig. 3 does not add to the results.

In the end of each paragraph, please give a brief sentence on the main finding within.

Regarding the muscle parameters: Did you assess the mean fiber diameter next to the area? Please add a histogram of the mean fiber size in each group.

Why were there no physiological muscle parameters assessed in the serum, like CK, CK-MB...?

The results section needs to be substantially shortened.

Discussion

Please shorten and focus on the brief description as comparison of your main findings to the literature.

How was the group size calculated? Did you do an a priori power analysis to be sure to have a sufficient sample size?

How did the randomization process work?

How many researchers performed the physiological training and testing? How was made sure that these were comparable?

Author Response

Thank you for your careful review of the manuscript. We have worked to respond to each reviewer comment:

  1. Abstract: Please do not start with what kind of model you have, but with a general overview of the field you are working in.

Response: Modified to further clarify that the field that we are working is determining the effectiveness of manual therapy in reducing the catabolic effects of performing repetitive intensive force tasks on bones.

  1. The first paragraphs of the introduction should rather focus on what is already known on manual therapy and its impact on bone metabolism instead of referencing your own works.

Response: Now began with an overview of musculoskeletal disorders from occupational overuse. Changed to clarify, as stated later in the introduction and discussion that the effects of manual therapy and its impact on bone catabolic changes occurring after repetitive intensive force tasks have not been studied. Also, explained more clearly that very few papers have examined the impact of manual therapy on bone metabolism and that this is the first paper to examine effects of manual therapy on bone in mature animals. 

Results

  1. The title of the first paragraph should not include a reference to the figure.

Response: Added a paragraph return at the end of the title of this first paragraph.

  1. Where is the main text to 2.1?

Response: Moved from Figure 1’s legend to the 2.1 section.

  1. Why was the radius chosen?

Response: The reason was previously described in the discussion. This reason has now been moved to section 2.1. The reason is: The radial bone was examined here since we have previously shown in this model that the radius is more affected than the ulnar bone [5, 6], and distal forearm bones are more affected by this task than more proximal upper extremity bones (such as, the humerus and scapula) [4].

  1. Tab. 1: Rather than the p-values, the means and standard deviation should be shown.

Response: The means and standard deviation are already shown in the figures (with the mean being height of the bar and the standard deviation being the spread of the dots). We have provided % 95 confidence intervals are also shown in the figures, since that is a stronger statistic than standard deviation or SEM. Adding that same information to the table would be redundant. Instead, Table 1 provides additional information regarding the two-way ANOVA or Mixed Model results (and for full disclosure of our results and their strengths).

  1. Without further explanation or verification, Fig. 3 does not add to the results.

Response: Figure 3 has been merged into Figure 2 since it is typical to show such representative figures in bone papers, in support of the graphical data. Removal would weaken the results shown in Figure 2. We altered the text further to add:Figure 2H shows representative 3D transaxial and coronal models of the distal radial trabeculae from each group that match the graphical results shown in Figure 2A-G.” See lines 141-142.

  1. In the end of each paragraph, please give a brief sentence on the main finding within.

Response: Added, except for lines 271 and 272 which already stated: “Overall, the total volume of loading (total reaches x total msec of grasping x the mean force per grasp) was higher in the TASK+MT rats, relative to the TASK rats.

  1. Regarding the muscle parameters: Did you assess the mean fiber diameter next to the area? Please add a histogram of the mean fiber size in each group.

Response: We clarified in the results in several places now that we assayed the cross-sectional area of the entire flexor muscle, not individual fiber sizes.  Individual myofiber size data is part of another manuscript and cannot be added to this manuscript. As indicated in the methods: “The widest circumference (in mm) of the flexor digitorum muscles was quantified (in mm2), from these cross-sections at 10x magnification.”

  1. Why were there no physiological muscle parameters assessed in the serum, like CK, CK-MB...?

Response: Since muscle inflammation is associated with reduced reach performance in this model [3, 12, 28], we focused on inflammation in this study. This was the correct choice in this case, since the number of macrophages numbers in flexor digitorum muscles correlated with pulling force and several key attributes of radial bone microstructure. Adding additional data would expand the results section further without adding critical information.

  1. The results section needs to be substantially shortened.

Response: Shortened where possible. Regrettably, adding a summary at the end of each paragraph as recommended expanded it somewhat.

Discussion

  1. Please shorten and focus on the brief description as comparison of your main findings to the literature.

Response: Reduced where possible. However, the girth of this study with results of task, preventive manual therapy, rest, and reversal manual therapy on bone requires discussion, as do the observed serum and muscle inflammatory changes.

  1. How was the group size calculated? Did you do an a priori power analysis to be sure to have a sufficient sample size?

Response: Added to lines 677-677. The sample size for this study was derived from our previous studies [20, 22], assuming a power of 80% and level of significance of 0.05, and a standard deviation of 28 for grip strength.

  1. How did the randomization process work?

Response: Thank you for this comment. We revised the methods to read on lines 475-477: “Rats were numbered and randomly assigned to groups by one group member (MA) at the beginning of the study to ensure blinding of tissue outcomes (operators providing the manual therapy treatments to specific rats could not be blinded; therefore, they were not engaged in the tissue analyses).” No blinding is possible for individuals responsible for training and task performance of TASK versus rats (so as to keep control rats from being trained) and no blinding is possible for the individuals responsible for the manual therapy treatment (so as to treat only the correct rats). Blinding was needed only for the analysis of collected tissues. That information was strengthened throughout the methods for each method.

  1. How many researchers performed the physiological training and testing? How was made sure that these were comparable?

Response: The researchers performing the physiological training and testing were the same for each test throughout the length of the study. We clarified further on lines 492-494, and 509-513 that the task performance and its outcomes are recorded continuously by a computer and Med PC Force Lever program during each task session and  that the data is later extracted from the program data into executable file Excel spreadsheets. Such computerized methods removes user bias and provides comparable outcomes across groups and even years. With regard to the manual therapy, as stated already on lines 527, the individuals providing the manual therapy treatments were the same as in previous studies (Barbe, Harris, et al. 2021; Barbe, Panibatla, et al. 2021). As stated on lines 586-588, microCT analysis was performed by an individual who was blinded to group assignment. It is also performed using a computer and computer defined methods. We strengthened for each method (bone and muscle tissue histomorphometry), that coded slides were used to mask group assignment and that one person is engaged in each assay only, and that computerized image analysis systems and methods are used to provide consistency.

Reviewer 2 Report

This manuscript researched the effect of manual therapy on the bone microstructural decline by using a rat model of the repetitive forceful task. The authors developed an interesting rat model of upper extremity overuse injuries and evaluated the preventive effect of manual therapy on bone microstructural declines. This topic is practical and useful. However, some issues should be addressed before final publication.

Firstly, some abbreviations are not correct, like CTX1, TNFalpha. CTX-1, TNF-α would be more formal. In addition, if the abbreviation is first used in the manuscript, please give your explanation. For instance, TASK, MT etc.

It would be beneficial to the readers if the author could detail the manual therapy in the introduction. Actually, I did not understand what manual therapy is when I was reading it for the first time.

At last, please add the graph legends to the figure if one graph contains several groups.

Author Response

  1. This manuscript researched the effect of manual therapy on the bone microstructural decline by using a rat model of the repetitive forceful task. The authors developed an interesting rat model of upper extremity overuse injuries and evaluated the preventive effect of manual therapy on bone microstructural declines. This topic is practical and useful. However, some issues should be addressed before final publication.

Response: Thank you for your kind and supportive words.

  1. Firstly, some abbreviations are not correct, like CTX1, TNFalpha. CTX-1, TNF-α would be more formal. In addition, if the abbreviation is first used in the manuscript, please give your explanation. For instance, TASK, MT etc.

Response: CTX-1 was already used. We changed to text to TNF-α, except for in citation titles (we kept previous publication titles).

  1. It would be beneficial to the readers if the author could detail the manual therapy in the introduction. Actually, I did not understand what manual therapy is when I was reading it for the first time.

 Response: A description has now been added to the introduction and matches what is described and referenced in the materials and methods, lines 520-526: The treatment was based on a previously developed manual therapy protocol performed on unsedated rats (Bove et al. 2016). It included gentle forearm tissue mobilization, forearm skin rolling i.e., a treatment intended to emulate "myofascial release" or "muscle stripping," to the forearm flexor compartment, mobilization of the wrist joints, and a gentle traction (stretch and glide) to the entire upper extremity. We also added direct manipulation of the palm in this study, where the tip of the rat’s index finger was pressed into the palm, at and just distal to the transverse carpal ligament, with a rolling motion.”

  1. At last, please add the graph legends to the figure if one graph contains several groups.

Response: Reviewer 1 has asked for shortening the results section. Since group names are now defined in section 2.1, and in Figure 2’s legend, we will keep with the norm to define group abbreviations on first use.

Round 2

Reviewer 1 Report

All requests have been fulfilled.

Please let the whole manuscript be spell-checked by a native speaker.